# Low-defect-density WS$_2$ by hydroxide vapor phase deposition

Yi Wan[1,2], En Li[3], Zhihao Yu[4,5], Jing-Kai Huang [6], Ming-Yang Li [4], Ang-Sheng Chou[4], Yi-Te Lee[7], Chien-Ju Lee[7], Hung-Chang Hsu[8], Qin Zhan[9], Areej Aljarb[10], Jui-Han Fu[1,11], Shao-Pin Chiu[7], Xinran Wang [5], Juhn-Jong Lin [7], Ya-Ping Chiu [8], Wen-Hao Chang [7,12], Han Wang[4], Yumeng Shi [13], Nian Lin [3], Yingchun Cheng [9✉], Vincent Tung [1,11✉] & Lain-Jong Li [2✉]

Two-dimensional (2D) semiconducting monolayers such as transition metal dichalcogenides (TMDs) are promising channel materials to extend Moore's Law in advanced electronics. Synthetic TMD layers from chemical vapor deposition (CVD) are scalable for fabrication but notorious for their high defect densities. Therefore, innovative endeavors on growth reaction to enhance their quality are urgently needed. Here, we report that the hydroxide W species, an extremely pure vapor phase metal precursor form, is very efficient for sulfurization, leading to about one order of magnitude lower defect density compared to those from conventional CVD methods. The field-effect transistor (FET) devices based on the proposed growth reach a peak electron mobility ~200 cm$^2$/Vs (~800 cm$^2$/Vs) at room temperature (15 K), comparable to those from exfoliated flakes. The FET device with a channel length of 100 nm displays a high on-state current of ~400 μA/μm, encouraging the industrialization of 2D materials.

[1] Physical Sciences and Engineering Division, King Abdullah University of Science and Technology (KAUST), Thuwal, Kingdom of Saudi Arabia. [2] Department of Mechanical Engineering, The University of Hong Kong, Hong Kong, China. [3] Department of Physics, The Hong Kong University of Science and Technology, Hong Kong, China. [4] Corporate Research, Taiwan Semiconductor Manufacturing Company (TSMC), Hsinchu, Taiwan. [5] National Laboratory of Solid State Microstructures, School of Electronic Science and Engineering and Collaborative Innovation Center of Advanced Microstructures, Nanjing University, Nanjing, China. [6] School of Materials Science and Engineering, University of New South Wales, Sydney, NSW, Australia. [7] Department of Electrophysics, National Yang Ming Chiao Tung University, Hsinchu, Taiwan. [8] Department of Physics, National Taiwan University, Taipei, Taiwan. [9] Key Laboratory of Flexible Electronics & Institute of Advanced Materials, Nanjing Tech University, Nanjing, China. [10] Department of Physics, King Abdulaziz University (KAAU), Jeddah, Saudi Arabia. [11] Department of Chemical System and Engineering, School of Engineering, The University of Tokyo, Tokyo, Japan. [12] Research Center for Applied Sciences, Academia Sinica, Taipei, Taiwan. [13] School of Electronics and Information Engineering, Shenzhen University, Shenzhen, China. ✉email: iamyccheng@njtech.edu.cn; vincent.tung@kaust.edu.sa; lanceli1@hku.hk

For high-performance electronics in advanced technology nodes, the thickness of transistor channels needs to be as thin as possible to ensure sufficient gate control with the gate length scaling[1]. Therefore, the transition metal dichalcogenide (TMD) monolayer around 1 nm thick has been considered as a promising channel material for future nodes[2,3]. The extraordinary properties of atomically thin 2D TMDs are profoundly influenced by the presence of imperfections[4,5]. It has been widely accepted that the electrical quality of mechanically exfoliated TMD monolayer flakes is superior to those from synthetic processes[4,6]; however, the non-scalability impedes their practical applications. TMD monolayers from scalable synthetic approaches like chemical vapor deposition (CVD) method usually contain abundant of imperfections including grain boundaries, point defects and strain[5,6]. Recently, oxygen substituted sulfur vacancy ($O_S$) has been demonstrated as the dominant point defect in CVD samples by scanning tunneling microscope (STM) measurements[7]. Although some efforts have been made to reduce the point defects, for example, thiol chemistry[8] and chalcogen gas annealing[9] for 'repairing' the chalcogen vacancy, it remains a formidable challenge to passivate other substitutional point defects. Hence, minimizing the defect density of synthetic 2D TMDs is crucial for achieving high electronic properties for practical applications.

Among the typical TMD monolayers, $WS_2$ exhibits high mobilities and saturation velocities for both electrons and holes based on the full-band Monte Carlo analysis of the Boltzmann transport equation[10,11]. Conventional CVD methods can provide scalable $WS_2$ monolayers through the direct sulfidation of either tungsten trioxide ($WO_3$) or other oxygen-containing precursors[12–14]. Although single crystal $WS_2$ flakes can be achieved, abundant defects are still present[15,16], which in turn leads to insufficient performance for advanced electronic devices. Transport agents like water[17–19] and oxygen[20,21] have been used to enhance the volatilization of metal source for improving the growth; however, their impact on materials have seldom been explored.

In this work, we discover that hydroxide vapor phase deposition (OHVPD) enables the growth of $WS_2$ monolayers with a significantly lower density of structural defects. The simulation results prove that W-OH bond in the hydroxide intermediates provides an energy favorable route for the sulfurization process. By analyzing the statistical photoluminescence (PL) and Raman results, OHVPD-$WS_2$ shows superior optical quality compared to conventional CVD-$WS_2$. STM measurements for the OHVPD-$WS_2$ monolayers transferred onto conducting substrates present the total defect density in the order of $10^{12}$ cm$^{-2}$ which is one order magnitude lower than that of CVD-$WS_2$. As-grown low-defect-density $WS_2$ monolayer show prominent electrical performance including high electron mobility of ~200 cm$^2$/Vs (~800 cm$^2$/Vs) at room temperature (15 K), and high current density of ~400 μA/μm for short channel device.

## Results and discussion

### Hydroxide vapor phase deposition for $WS_2$ monolayers. 
In contrast to the direct sulfidation of $WO_3$, the OHVPD method utilizes water vapors to transport high-purity W metal to reduce the incorporation of oxygen and other impurities (such as Mo atoms in the $WO_3$ source) into the deposited $WS_2$ films, where the growth is schematically depicted in Fig. 1a. The W metal undergoes a few oxidation steps with water vapors to form tungsten hydroxide $WO_2(OH)_2$ at an elevated temperature[22–24] as evidenced by the X-ray Diffraction (XRD) results (see Supplementary Note 1 and Supplementary Figs. 1 and 2 for details). The volatile $WO_2(OH)_2$ intermediates transported to the target substrates at the downstream area are reduced in the presence of sulfur vapors and $H_2$ gases to form $WS_2$ monolayer crystals. Although $WO_2(OH)_2$ and conventionally used $WO_3$ may undergo similar reduction paths to form $WS_2$, the sulfidation kinetics is distinctly different in both cases[25,26]. We construct molecular models to understand the difference in sulfurization of the oxygen and hydroxide bonded in $WS_2$ (details in Supplementary Fig. 3) and the key reaction step is depicted in Fig. 1b. Our simulation shows that the W-O bond length of the bonded oxygen W-O (2.061 Å) is shorter than that of the bonded hydroxide W-OH (2.152 Å), and the kinetic barrier for breaking W-O (1.440 eV) is higher than the 0.936 eV for W-OH. Also, it needs two H atoms to form $H_2O$ for the removal of the bonded O, while only one H atom is required for the bonded hydroxide. The results indicate that the presence of W-OH bonds provides a more energy favorable route to perform the sulfidation[26]. Figure 1c, d shows the typical optical and atomic force microscopy (AFM) images of the OHVPD-grown $WS_2$ monolayers. Their domain size can reach several microns and the inch-scale continuous $WS_2$ monolayer film is also achievable (Fig. 1e). PL and Raman mapping results in Supplementary Fig. 4 present a homogeneous and high-quality OHVPD-$WS_2$ film.

### High optical qualities of OHVPD-$WS_2$ monolayers. 
Raman spectra of the as-grown $WS_2$ monolayers prepared by conventional sulfidation of $WO_3$ (labeled as CVD) and the proposed OHVPD are compared in Fig. 2a, where many characteristic modes are identified, including the in-plane vibration mode $E_{2g}^1(\Gamma)$ (~354 cm$^{-1}$), two defect-sensitive modes[27] out-of-plane $A_{1g}$ (~416 cm$^{-1}$) and longitudinal acoustic at **M** point in the Brillouin zone $LA(\mathbf{M})$ (~173 cm$^{-1}$), and others[28]. To qualitatively compare the defect level, 50 Raman spectra from various single crystals were collected for each type of samples. Figure 2b shows that the statistical average of $A_{1g}$ peak width of 5.5 cm$^{-1}$ for CVD-$WS_2$ is broader than the 4.2 cm$^{-1}$ for OHVPD-$WS_2$; meanwhile, the normalized intensity of $LA(\mathbf{M})$ peak of CVD-$WS_2$ is larger than that of OHVPD-$WS_2$ (Fig. 2c). These results indicate that OHVPD-$WS_2$ exhibits superior quality[27,29]. The room temperature photoluminescence (PL) spectra of monolayer OHVPD-$WS_2$ typically exhibit a higher peak energy and a narrower full width at half maximum (FWHM) compared with the CVD-$WS_2$ (see Supplementary Fig. 5 for details), indicating its superior quality[30]. Figure 2d shows the PL measurement for both samples at 4 K can be better deconvoluted by Gaussian functions, where the high energy mode is assigned to neutral exciton ($X^0$), the peak with a lower energy by ~30 meV is assigned to trion ($X^T$), and the broad peak with the lowest energy is assigned to defect-bound exciton ($X^D$). The significantly lower intensity of $X^T$ and $X^D$ peaks for OHVPD-$WS_2$ corroborate a lower defect density on its basal plane. We also applied OHVPD for $MoS_2$ growth; similarly, OHVPD-$MoS_2$ shows a higher peak energy and narrower FWHM compared with CVD-$MoS_2$ (see Supplementary Fig. 6).

### Defect analysis of $WS_2$ monolayers by STM. 
To investigate the structural defects in as-grown $WS_2$ monolayers, we perform scanning tunneling microscopy (STM) measurements to characterize their types and densities following Schuler et al.[7]. Figure 3a, b shows the STM images of CVD-$WS_2$ and OHVPD-$WS_2$ monolayers directly grown on conductive highly oriented pyrolytic graphite (HOPG) substrates. It is conspicuous that CVD-$WS_2$ exhibits a larger number of structural defects than OHVPD-$WS_2$. These typically observed defects can be categorized into a few types, including oxygen substituting upper ($O_{S(top)}$) and bottom sulfur ($O_{S(bottom)}$), molybdenum substituting tungsten

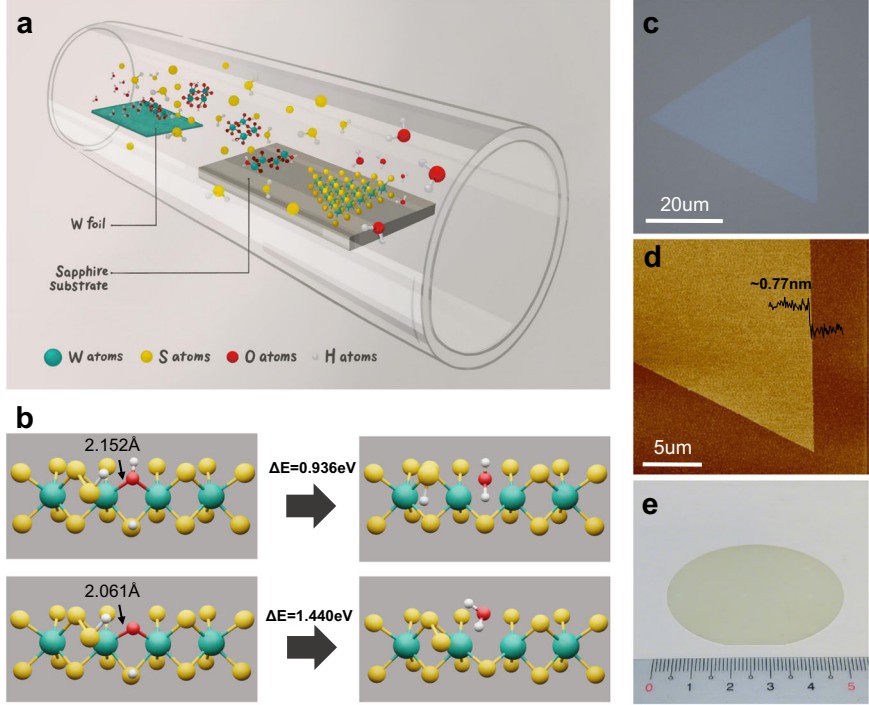

**Fig. 1 Hydroxide Vapor Phase Deposition. a** Schematic of hydroxide vapor phase deposition (OHVPD) growth of $WS_2$ monolayers. **b** Nudged elastic band (NEB) simulation of kinetic energy barriers ($\Delta E$) for bonded OH and O dissociating from the edge of $WS_2$. **c, d** Optical Image **c** and AFM image **d** of the OHVPD-$WS_2$ monolayer. **e** Photo of a 2-inch OHVPD-$WS_2$ monolayer film grown on a sapphire substrate.

($Mo_W$), other negatively charged defects (NCD), and positively charged defects (PCD), as featured in the magnified STM images in Fig. 3c–g. Only a very small number of sulfur vacancies are found (Supplementary Fig. 7). To estimate the area number density of various defects, at least more than 20 STM images (40 nm by 40 nm) for each CVD-$WS_2$ and OHVPD-$WS_2$ are analyzed, and the estimated densities of different structural defects are shown in Fig. 3h. It is noteworthy that we do not use high-resolution scanning transmission electron microscopy for quantitative characterization of defects owing to the potential damages by electron beams during the imaging and its difficulty in distinguishing $O_S$ from S-vacancy.

The $O_{S(top)}$ and $O_{S(bottom)}$ in CVD-$WS_2$ are at a similar density level, estimated as $3.52 \times 10^{12}\,cm^{-2}$ and $3.46 \times 10^{12}\,cm^{-2}$, respectively. These predominant defects are significantly lower in OHVPD-$WS_2$, $1.2 \times 10^{12}\,cm^{-2}$ for $O_{S(top)}$ and $1.18 \times 10^{12}\,cm^{-2}$ for $O_{S(bottom)}$. These major defects, including $O_{S(top)}$ and $O_{S(bottom)}$, do not introduce in-gap charged states, and their measured $dI/dV$ spectrums (See Supplementary Fig. 8) are close to that in pristine $WS_2$ regions, agreeing well with previous reports[7,31,32]. We also observe another neutral defect $Mo_W$ with the density in the order of $10^{12}\,cm^{-2}$ in CVD-$WS_2$, which is likely caused by the presence of Mo impurity in the W-precursors (~6.5ppm in $WO_3$ according to the material provider). Recent DFT simulation argues that $O_S$ does not introduce in-gap charged states and only marginally affects $WS_2$ electronic structures owing to the isoelectronic feature of S and O[7]. Also, the band structure of $WS_2$ with a $Mo_W$ closely resembles that of pristine $WS_2$[7]. Hence, we suspect that the electron mobility of $WS_2$ may not be critically affected by these neutral $O_S$ and $Mo_W$ defects in particularly at the density level lower than $10^{13}\,cm^{-2}$. We note that other recent reports have suggested that electron mobility in $MoS_2$ may be increased with the band gap narrowing effects from the incorporation of high density charged S vacancies[33] (up to $10^{14}\,cm^{-2}$) or with the screening effect by heavy oxygen doping[34], where these approaches are different from the

low-defect density requirement for scalable electronics. Charged defects scatter carriers through Coulomb interaction that can also lead to significant band bending and possibly a local potential change around the defects[7]. A recent report by Yu et al. has demonstrated the electron mobility of $MoS_2$ monolayer can be significantly enhanced by the passivation of charged S-vacancies using thiol molecules[8]. Therefore, the number of charge defects should be minimized as possible. Our STM results show that the density of NCDs in CVD-$WS_2$ ($3.9 \times 10^{10}\,cm^{-2}$) is almost five times of that in OHVPD-$WS_2$ ($8 \times 10^9\,cm^{-2}$). The measured $dI/dV$ spectra for NCDs (See Supplementary Fig. 9) is consistent with the reference[30] and the NCDs can be assigned as the S vacancies substituted with CH, C, or N atoms.

For device fabrication, the $WS_2$ monolayers typically need to be transferred from the sapphire growth substrates onto target substrates. To estimate the defect levels of $WS_2$ after the mechanical transfer processes, both samples are transferred from sapphire substrates onto HOPG substrates for STM analysis, shown as t-CVD and t-OHVPD in Fig. 3h. In this study, we adopt the polydimethylsiloxane (PDMS)-assisted transfer method (See methods)[35]. The overall defect density of t-OHVPD $WS_2$ ($2.5 \times 10^{12}\,cm^{-2}$) is one order of magnitude lower than that in t-CVD ($2.1 \times 10^{13}\,cm^{-2}$). Also, the total charge impurity in t-OHVPD $WS_2$ ($2.0 \times 10^{10}\,cm^{-2}$) is roughly one order of magnitude lower than t-CVD ($2.5 \times 10^{11}\,cm^{-2}$).

Interestingly, the density of $O_{S(top)}$ is much larger than $O_{S(bottom)}$ in t-CVD $WS_2$, which could be due to the differences in growth substrate (sapphire rather than HOPG). Similar features have also been observed in other systems, such as $WS_2$ on graphitized-SiC substrates[7,31] ($O_{S\ (bottom)} > O_{S(top)}$) and oxygen-doped $MoS_2$ on HOPG ($O_{S(top)} > O_{S(bottom)}$)[34]. The remarkably larger $O_{S(top)}$ than $O_{S(bottom)}$ and the increase in NCDs for t-CVD samples indicate that the $WS_2$ monolayers grown by conventional CVD are prone to damage from the subsequent transfer processes. It is suspected that the substitution

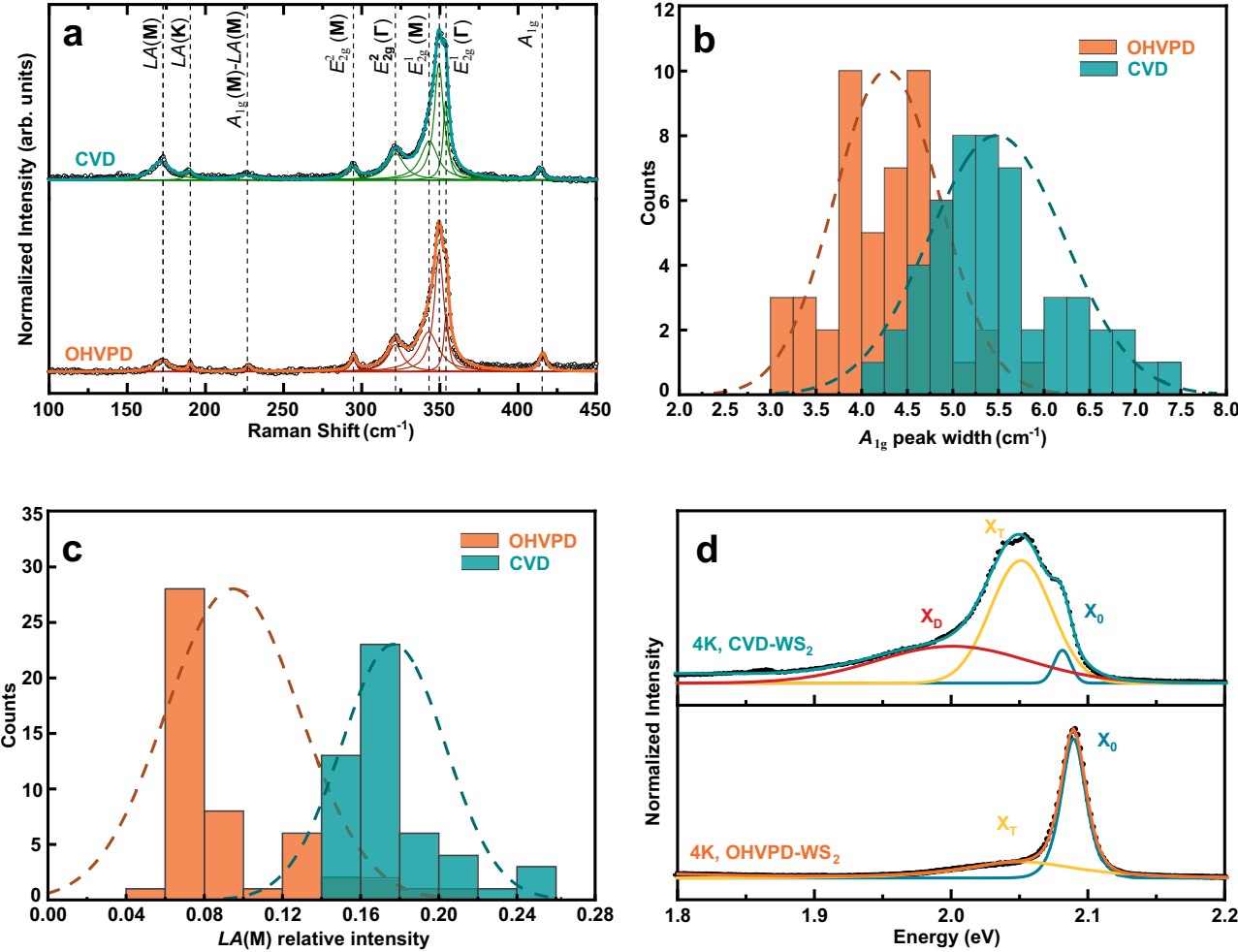

**Fig. 2 Optical characterizations of OHVPD- and CVD-WS₂ monolayers. a** Typical Raman spectra showing the characteristic modes of OHVPD- and CVD-WS₂ monolayers excited by 532 nm wavelengths. The hollow circles and coloured lines are the experimental and Lorentzian fit curves respectively. **b, c** Statistic distribution of out-of-plane mode $A_{1g}$ Raman peak width and normalized intensity of longitudinal acoustic at M point in the Brillouin zone $LA(\mathbf{M})$ Raman peak for OHVPD- and CVD-WS₂ monolayers. The dashed lines represent the normal distribution curves. **d** Low-temperature PL spectra of OHVPD- and CVD- WS₂ monolayers at 4 K. The solid lines and dashed ones are the experimental and fitted peaks respectively. The fitted peaks can be assigned to neutral exciton (X⁰), trion (Xᵀ), and defect-bound exciton (Xᴰ).

of S atoms by environmental oxygen to form neutral $O_{S(top)}$ defects and the reaction carbon or nitrogen impurity species to form NCDs occur during the transfer. Our simulation (Supplementary Fig. 10) suggests that the removal of S atoms adjacent to the $O_S$ on the same side tends to be thermodynamically and kinetically favorable by oxidation, agreeing with the observation that further O substitution is easier in the sample with a higher density $O_S$ (i.e., CVD samples). It is anticipated that top-side S atoms exposed to the chemicals and air should be oxidized easier compared to the bottom side. Thus, the $O_{S(top)}$ density is larger than the $O_{S(bottom)}$ density after transfer as revealed by the experiments. Hence, the growth of low-defect-density TMD films and the development of mild transfer methods warrant intense efforts and should be the focus of 2D electronics.

**Electrical performance of OHVPD-WS₂ monolayers.** For evaluating the electrical performance of the low-defect OHVPD-WS₂ monolayers, we fabricated field-effect transistors with the back-gate configuration and characterized their electrical properties in a high vacuum (∼ $10^{-6}$ Torr) using a standard four-probe technique. Figure 4a presents the four-probe conductivity $\sigma = (I_d/\Delta V) \times (L_{CH}/W_{CH})$ as a function of back-gate voltage $V_g$ at

various temperatures, where $I_d$ is the source-drain current; $\Delta V$, $L_{CH}$, and $W_{CH}$ are the voltage difference, length, and width between the two voltage probes, respectively. The OHVPD-WS₂ sample exhibits at least 10X higher conductivity than the typical CVD-WS₂ (in Supplementary Fig. 11). The OHVPD-WS₂ shows an apparent metal–insulator transition (MIT) at around $V_g = 60$ V (corresponding to the carrier density $n = C_{OX}V_g \sim 4.3 \times 10^{12}$ cm⁻²), where $C_{OX}$ ($1.15 \times 10^{-8}$ F cm⁻²) is the geometric gate capacitance per unit area for a 300 nm SiO₂ dielectric. The MIT has been observed in the low charge-trap-state sample, i.e., as-exfoliated or vacancy-passivated samples[8,36,37]. Using the model proposed in reference[38], we estimate the trap density ($N_{tr}$) and charge impurity density ($N_{CI}$) in OHVPD-WS₂ as ∼$3.6 \times 10^{12}$ cm⁻² and ∼$8.7 \times 10^{10}$ cm⁻² (see Methods and Supplementary Fig. 12), which are the lowest compared to the reported MoS₂ and WS₂ monolayers (see Supplementary Fig. 13 and Supplementary Table 1). The extracted $N_{CI}$ is around four times of the charge defects observed by STM (∼$2 \times 10^{10}$ cm⁻² for t-OHVPD in Fig. 2h), suggesting that some $N_{CI}$ may come from other sources such as the WS₂-SiO₂ interfaces and e-beam lithographic processes during the metal line patterning. We have also extracted the $N_{tr}$ and $N_{CI}$ for CVD-WS₂ as ∼ $8.2 \times 10^{12}$ cm⁻² and

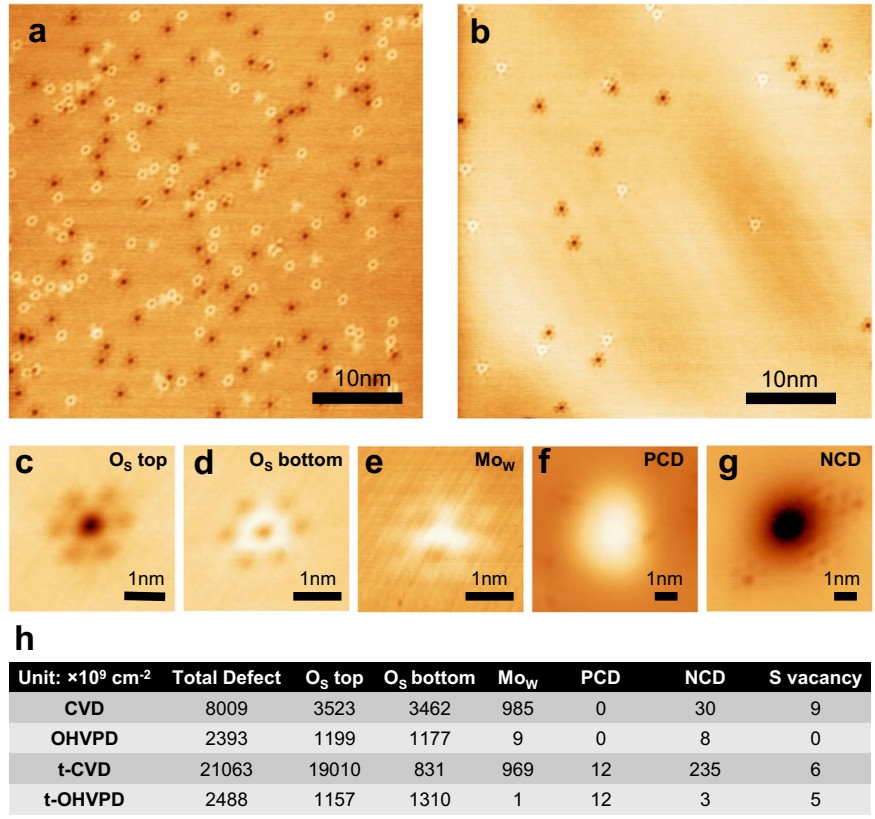

**Fig. 3 Defect analysis by scanning tunneling microscopy (STM). a**, **b** STM images of **a** CVD-WS$_2$ (Bias Voltage (V) = 1.35 V, Current (I) = 40 pA) and **b** OHVPD-WS$_2$ monolayer (V = 1.15 V, I = 30 pA). **c–f** STM images (V = 1.1 V, I = 30 pA) of the commonly observed point defects in CVD- and OHVPD-WS$_2$: oxygen substituting sulfur (O$_s$) in the **c** top and **d** bottom sulfur plane; **e** Mo substitutional tungsten (Mo$_W$); **f** Positively charged defect (PCD) and **g** Negatively charged defect (NCD). **h**, Histograms table of observed point defect density in different OHVPD- and CVD- WS$_2$.

| Unit: ×10⁹ cm⁻² | Total Defect | O$_s$ top | O$_s$ bottom | Mo$_W$ | PCD | NCD | S vacancy |
|---|---|---|---|---|---|---|---|
| **CVD** | 8009 | 3523 | 3462 | 985 | 0 | 30 | 9 |
| **OHVPD** | 2393 | 1199 | 1177 | 9 | 0 | 8 | 0 |
| **t-CVD** | 21063 | 19010 | 831 | 969 | 12 | 235 | 6 |
| **t-OHVPD** | 2488 | 1157 | 1310 | 1 | 12 | 3 | 5 |

~2.2 × 10$^{12}$ cm$^{-2}$ (based on the results in Supplementary Fig. 12), where the trap density (from both the interface and channel defects) is comparable to the OHVPD-WS$_2$. However, the extracted $N_{CI}$ is ~ 25 times higher than that in OHVPD-WS$_2$. Note that the extracted $N_{CI}$ for CVD-WS$_2$ (2.2 × 10$^{12}$ cm$^{-2}$) is much higher than the STM charge defect density of 2.5 × 10$^{11}$ cm$^{-2}$, suggesting that the WS$_2$ with more structural defects may incur more charge impurities during the device fabrication processes. Supplementary Fig. 14 shows the typical dual-sweep transfer curves of our devices. The normalized hysteresis width is 40 mV/MV cm$^{-1}$, which is on par with reported hysteresis values and indicates the presence of low border traps and interface states[39].

We adopt the expression $\mu_{FE} = (d\sigma/dV_g) \times (1/C_{OX})$ to extract the field-effect mobility $\mu_{FE}$ for OHVPD-WS$_2$ in four-probe measurements (at the carrier concentration of $n = \sim 4.7 \times 10^{12}$ cm$^{-2}$) and the $\mu_{FE}$ reaches 198 cm$^2$V$^{-1}$s$^{-1}$ (789 cm$^2$V$^{-1}$ s$^{-1}$) at room temperature (15 K) as shown in Fig. 4b, recognized as the highest value among the reported synthetic monolayer WS$_2$. The $\mu_{FE}$ for CVD-WS$_2$ is significantly lower, ~ 17 cm$^2$ V$^{-1}$ s$^{-1}$ (105 cm$^2$ V$^{-1}$ s$^{-1}$) at room temperature (15 K). Figure 4c and Supplementary Fig. 15 compare the statistical results of electron mobility for the transistors based on OHVPD-WS$_2$, and CVD-WS$_2$ and exfoliated WS$_2$ from literature[38,40–43], where the electron mobility of OHVPD-WS$_2$ is comparable to the exfoliated WS$_2$ but obviously superior to CVD-WS$_2$. In addition, Fig. 4d demonstrates that the short-gate-length FET based on OHVPD-WS$_2$ monolayer can reach a maximum $I_{on} \approx 403$ μA/μm and $I_{on}/I_{off}$ current ratio ~ 10$^8$ at $V_{ds} = 1$ V, significantly higher than that made from CVD-WS$_2$ monolayer using the same device fabrication processes. Supplementary Fig. 16 shows the output characteristics of this short-gate-

length FET, which demonstrates promising current control and saturation. These facts point out that the density of charged defects is a critical factor that limits the performance of 2D monolayers. In conclusion, the as-grown CVD films with higher defect densities are susceptible to transfer and device fabrication processes. Our proposed OHVPD approach provides a route to largely reduce the defects directly from growth, which makes synthetic 2D TMDs potential for electronic applications.

## Methods

**Materials synthesis and transfer.** CVD-WS$_2$ monolayer samples were synthesized on sapphire substrates by the typical CVD method with tungsten oxide (WO$_3$, Sigma-Aldrich, 99.995%) powders and sulfur (S, Sigma-Aldrich, 99.99%) powders as precursors. Generally, the S powders, WO$_3$ powders and sapphire substrates were placed on the upper stream, center and downstream of the furnace, respectively. After the chamber pressure went down to 1mtorr, Ar/H$_2$ gas was purged into the chamber and kept the chamber pressure at 10 torr. The temperature was elevated to 900 °C and kept for 15 min for growth.

OHVPD-WS$_2$ monolayer samples were achieved in a homemade 3-inch CVD system. High purity tungsten foil (W, 99.95%) and S powders were used as precursors. Moisture was delivered into the growth chamber by Ar gas flow (180 s.c.c.m.) at atmospheric pressure. The S powders were placed upstream of the tube and the temperature was controlled by an additional heating belt at 180 °C. The W foil was placed on the center of the furnace at 1050 °C while the sapphire substrates were placed downstream at 950-800 °C. During the growth, H$_2$ gas (20 s.c.c.m.) was delivered into the chamber for the reaction. The growth was kept for 15 min and followed by natural cooling to room temperature with the same carrier gas (Ar/H$_2$ 180/20 s.c.c.m.) without the presence of water vapors. More details on the growth process and results are provided in Supplementary Note 3 and Supplementary Figs. 17–18.

The as-grown monolayer WS$_2$ samples were transferred onto the target substrates by a polydimethylsiloxane (PDMS)-assisted transfer method[35]. The PDMS and hardener were mixed at a ratio of 10:1 (Sungyoung, PDMS 184 AB) in a clean beaker. The mixed solution was poured into the plastic container until 2 mm in thickness. The container was then put in a vacuum chamber for 2 h to remove

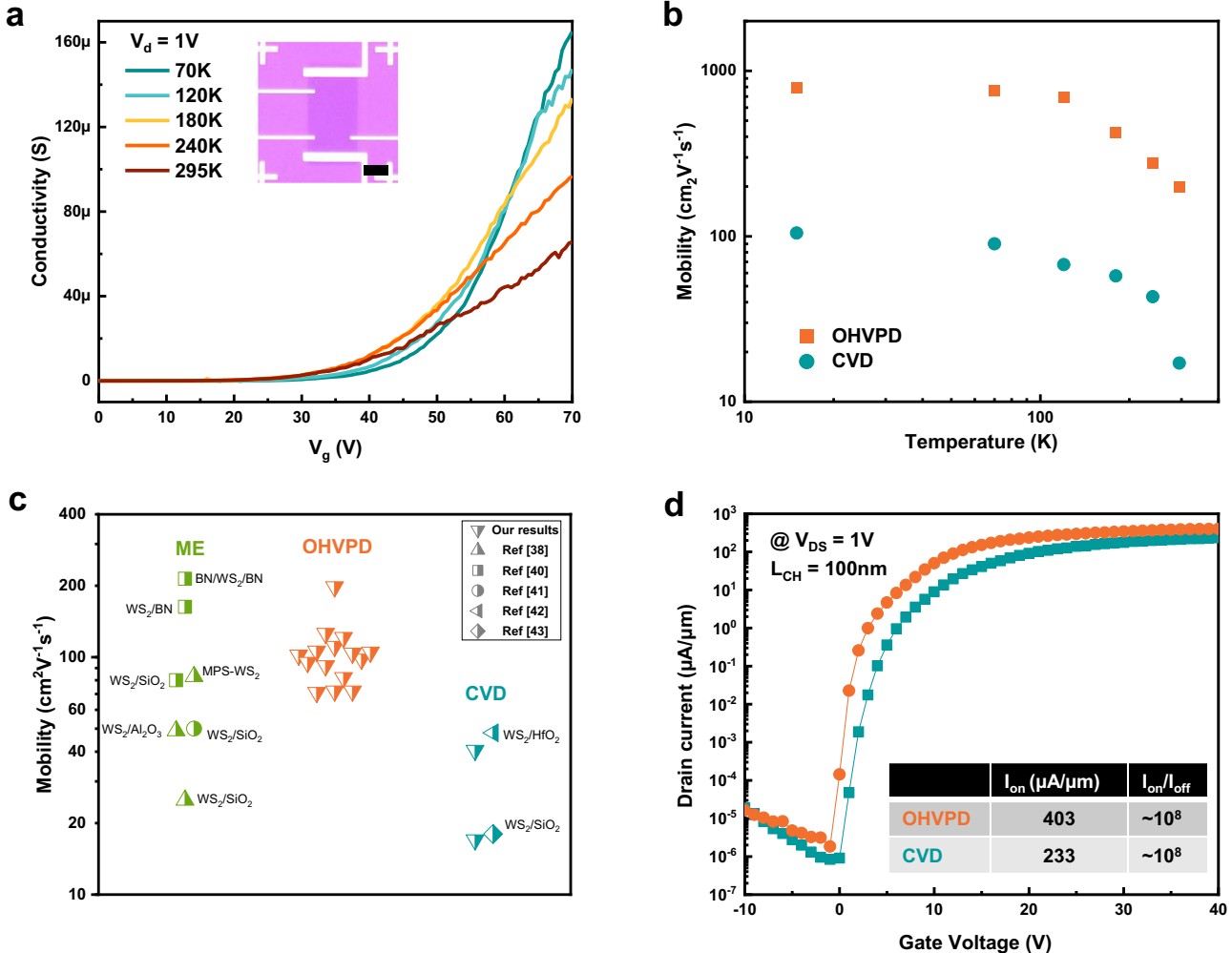

**Fig. 4 Electrical performance of OHVPD-WS$_2$ monolayers. a** Four-probe conductivity as a function of Vg for OHVPD-WS$_2$ monolayer device on the 300 nm SiO$_x$ substrate at different temperatures. Insect shows the device structure. (Scale bar: 5 μm) **b** Field-effect mobility as a function of temperature for OHVPD- and CVD-WS$_2$ monolayers. **c** Comparison of mobility distribution for our OHVPD-WS$_2$ results (orange), mechanical exfoliation WS$_2$ monolayers (ME, green), and conventional CVD-WS$_2$ (cyan) from literatures. **d** FET transfer curve of an OHVPD-WS$_2$ monolayer for the short channel device (L$_{CH}$ = 100 nm).

the bubbles and the PDMS film was cured at 50 °C for 24 h. To perform the transfer, the PDMS film was smoothly placed on the top of as-grown WS$_2$ on sapphire and soaked the whole stacked films into 1 M KOH$_{(aq)}$ for 5 min to weaken the interaction between WS$_2$ and sapphire substrates. Next, the PDMS/WS$_2$ film was slowly peeled off from sapphire and rinsed with DI water to remove the residue. The sample was then transferred to the target substrate and annealed at 70 °C for 20 min to remove the residue water and increase the adhesion. Finally, the PDMS film was slowly peeled off to get a clean WS$_2$ sample on the target substrate.

**Device fabrication and electrical measurements.** For the short channel device, the monolayer WS$_2$ films were transferred onto the commercial SiN$_x$ film (thickness = 100 nm) on p$^{++}$-Si substrates as back gated field-effect transistors (FET). Then Helium-ion beam lithography (ORION NanoFab, Zeiss) with the ion-beam-resist, PMMA (Allresist, AR-P 672-Serie, spin-coated with 4000 rpm for 40 s and baked at 180 °C for 3 min.) was used to pattern the source/drain (S/D) metal contacts, which defined the channel length (L$_{CH}$) from 100 nm to 400 nm and was developed by using 1:3 mixture of 4-methyl-2-pentanone (MIBK) and isopropyl alcohol (IPA). For the contact metals, 20 nm of Bi followed by 15 nm of gold (Au) encapsulating layer were deposited on the WS$_2$ using e-gun evaporation at a high vacuum chamber (~1 × 10$^{-7}$ torr). The metal lift-off process was carried out in warm acetone (60 °C) and then rinsed by IPA. Finally, the WS$_2$ electrical characteristics were measured in a vacuum system (10$^{-5}$–10$^{-6}$ Torr) in a Lakeshore probe station using a Keithley 4200-SCS parameter analyzer.

For the four-terminal device measurement, the heavily doped Si substrate was used as a back gate and the 300 nm SiO$_2$ was used as a gating dielectric. The devices were patterned using PMMA masks and electron beam lithography. 5 nm Al and

65 nm Au electrodes were deposited using e-beam evaporation. The electrical characterization of monolayer WS$_2$ FETs was carried out under vacuum (<10$^{-4}$ Torr) in a JANIS CCS350 closed-cycle refrigerator (10–500 K). Our four-terminal measurements were performed from 15 to 300 K, starting from the lowest temperature. The gate and drain biases are provided by the Keithley Model K-6430 Sub-Femtoamp Remote Source Meters, which are also used to monitor the leakage current and drain current. And the Keithley Model K-2182 is used to sense the voltage difference as a voltmeter. In our structure, the voltage sensed probes minimally affect the current flow in the channel material and thus act like perfect voltmeters.

**Estimation of the N$_{tr}$ and N$_{CI}$ in monolayer WS$_2$.** The theoretical model we used was proposed by Wang's group[38]. In brief, the band mobility (the mobility for free carriers) of monolayer WS$_2$ was calculated according to Matthiessen's rule, which is expressed as $\mu_0(n, T)^{-1} = \mu_{ph}(T)^{-1} + \mu_{CI}(n, T)^{-1}$. Here we ignore phonon-limited mobility as a result that theoretical phonon-limited mobility is much higher than the experimental values over the entire temperature range. The CI-limited electron mobility $\mu_{CI}$ can be defied by

$$\mu_{CI} = \frac{2e}{n\pi\hbar^2 k_B T} \int_0^\infty f(E)[1 - f(E)]\Gamma_{CI}(E)^{-1}EdE$$

where $e$, $\hbar$, $k_B$, $T$ and $f(E)$ are the electron charge quantum, the Planck constant divided by 2π, the Boltzmann constant, the temperature and the Fermi-Dirac distribution, respectively. Moreover, the experimental mobility μ is not exactly equal to the band mobility $\mu_0$, due to the charge traps, which reduce the free carrier population and is responsible for the MIT. The density of conducting electrons in

the extended states can be calculated by

$$n_c(n, T) = \int_0^{+\infty} N_0 \frac{1}{e^{(E-E_F)/k_B T} + 1} dE$$

Finally, we can calculate the experimental mobility $\mu$ by

$$\mu(n, T) = \mu_0(n, T) \frac{\partial n_c(n, T)}{\partial n}$$

**STM measurement**. Our STM experiments were conducted in the commercial ultra-high vacuum LT-STM system (CreaTec) with a base pressure of $1.0 \times 10^{-10}$ mBar. All STM images were acquired at 77 K in the constant-current mode by using a chemically etched tungsten tip and the bias voltages refer to the sample with respect to the STM tip. Before measurement, the samples were annealed at ~550 K for over 3 h to remove possible adsorbates. Note that such a low annealing temperature is used to avoid the transition to sulfur vacancies[31]. The $dI/dV$ spectra were acquired at 5.3 K by using a lock-in technique with the bias modulation of 20 meV at 717.3 Hz.

**Kinetic simulation of sulfurization process**. It is important to understand the role of $H_2O$ in the formation of $WS_2$ monolayer during growth. Here, we applied nudged elastic band (NEB) method[44–46] to model the energy barrier of the sulfurization process with and without $H_2O$. Without $H_2O$, the precursor used in conventional CVD is $WO_3$. Therefore, there are W-O bonds at the edges (or the growth-front) of CVD-$WS_2$. On the other hand, the edge of OHVPD-$WS_2$ possibly contains W-OH bonds due to the $H_2O$ supply. A monolayer of $5 \times 4 \times 1$ supercell of $WS_2$ with the zigzag edge is used. The vacuum of 15 and 20 Å along y and z directions are applied to avoid interaction between their replica images because of periodic conditions. A gamma-centered $1 \times 1 \times 1$ k-mesh is employed for ion relaxation and NEB calculation. Supplementary Fig. 3a shows the kinetic energy barriers of transformation of W-OH to W-SH group. The three major barriers are 0.41, 0.94 and 0.71 eV, respectively. The first barrier is 0.41 eV, which is related to one H atom from $H_2S$ molecular to S atom near O atom at the $WS_2$ edge. The second barrier is 0.94 eV, which corresponds to the H on S transfer to O atom and then the formation of $H_2O$. The third barrier is 0.71 eV, relating to the detachment of $H_2O$ from the $WS_2$ edge. Supplementary Fig. 3b shows the kinetics of transformation of W-O to W-S group. There are also three barriers. The first barrier is 0.17 eV, which is related to the transfer of one H from $H_2S$ molecular to one S atom on $WS_2$ edge. However, the second and third barriers are 1.44 and 1.39 eV, respectively, which are much larger than the case of transformation from W-OH to W-SH group. The second barrier is related to the two H atoms moving to O at the edge. The third barrier corresponds to the leaving of $H_2O$ from edge of $WS_2$. The difference of energy barriers for two different scenarios indicates W-O bond is much more difficult in transforming to W-S group compared to the W-OH to W-SH bond.

**Simulation of $O_S$ defect formation**. It has been proposed that the reaction of $O_2$ with $WS_2$ is one origin of $O_S$ defect formation[47]. The formation energy for one $O_S$ defect generation in pristine $WS_2$ can be described as the following equation:

$$E^{form} = E^{def} - E^{pristine} - n\mu_O + mE^{SO2}$$

Where $E^{form}$ is the formation energy of the $O_S$ in $WS_2$, $E^{def}$ and $E^{pristine}$ are total energies of defective and pristine $WS_2$, $\mu_O$ is the chemical potential of oxygen, $E^{SO2}$ is the total energy of $SO_2$ gas molecular, n and m are the number of substitutional O and formed $SO_2$. A monolayer of $6 \times 6 \times 1$ supercell of $WS_2$ is used to investigate the oxidation process. A vacuum of 20 Å is applied to avoid interaction between their replica images because of periodic conditions. A gamma centered $2 \times 2 \times 1$ k-mesh is employed for ion relaxation and NEB calculation.

Supplementary Fig. 10a shows the formation energies of $O_S$ in $WS_2$. $O_S$ defects are thermodynamically favorable under O-rich conditions. Moreover, the more $O_S$ defects in $WS_2$, the much lower the formation energy. This indicates once more $O_S$ are formed, it is much easier to further have O substitution in the $WS_2$ system from a thermal dynamics point of view.

It is also vital to investigate the kinetics of $O_2$ dissociation and the formation of $O_S$ in $WS_2$. The NEB method was applied to model the energy barrier for the two steps. Different from the previously proposed two steps formation of $O_S$ in $MoS_2$[48], where the S vacancy on $MoS_2$ forms firstly, and then the O occupies the S vacancy position. Supplementary Fig. 10b shows the kinetics and energetics of the $O_2$ dissociation and formation of $O_S$ in the pristine $WS_2$. There are two major barriers. The first one is $O_2$ molecular absorption on top of the S atom in the $WS_2$ plane. The energy barrier is 1.20 eV, which is for the breaking of O–O bond. One O atom gets close to one W atom, and the other connects with one S atom. Such configuration is about 1.5 eV lower than the initial configuration. The second barrier is 1.075 eV for the break of one W–S bond, leading to the S atom lifted up. In the end, the SO group is out of the plane and can be taken by other $O_2$ or $H_2O$ molecules. The two barriers are over 1.0 eV, suggesting the oxidation process is extremely slow. Supplementary Fig. 10c–d shows the kinetics and energetics of the further O substitution step in the $WS_2$ with one $O_S$ defect at the flipside near $O_S$ and the same side near $O_S$. Both barriers are lowered compared with

Supplementary Fig. 10b, indicating the presence of $O_S$ defects will accelerate the oxidation process.

## Data availability

Relevant data supporting the key findings of this study are available within the article and the Supplementary Information file. All raw data generated during the current study are available from the corresponding authors upon request.

## Code availability

The code used to plot the data is available from the corresponding authors upon request.

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

## Acknowledgements

V.T., and Y.W. are indebted to the support from the King Abdullah University of Science and Technology (KAUST) Office of Sponsored Research (OSR) under Award No: OSR-2018-CARF/CCF-3079. E.L. and N.L. acknowledge the support of Hong Kong UGC (C6012-17E). H.W., M.Y.L., and A.S.C. thanks the support from Taiwan Semiconductor Manufacturing Company (TSMC). W.H.C. acknowledges the supports from the Ministry of Science and Technology of Taiwan (MOST-108-2119-M-009-011-MY3, MOST-107-2112-M-009-024-MY3) and from the CEFMS of NCTU supported by the Ministry of Education of Taiwan. L.J.L. and Y.W. acknowledge the support from the University of Hong Kong. Special thanks to Kate Chuang for her assistance.

## Author contributions

Y.W., L.J.L. and V.T. conceived the project. Y.W., J.-K.H. and M.-Y.L. performed the synthesis of CVD- and OHVPD-WS₂, and carried out Raman, PL, and AFM characterizations. Z.Y., A.-S.C., M.-Y.L., J.-J. L., S.-P.C., Y.-T.L. and X.W. conducted fabrication of field-effects transistors and associated calculations. E.L., H.-C.H., Y.-P.C. and N.L. performed and analyzed STM/STS characterization. C.-J.L. and W.-H.C. performed and analyzed the low-temperature PL measurement. Q.Z. and Y.C. performed the first-principles and nudged elastic band calculations. H.W., A.A., J.-H.F. and Y.S. provided constructive opinions and suggestions. All authors discussed and contributed the results. Y.W., L.J.L. and Y.C. wrote the paper.

## Competing interests

The authors declare no competing interests.
