## [Peer Review File · Nature Communications]

Low-defect-density WS₂ by hydroxide vapor phase depositionREVIEWER COMMENTS

Reviewer #1 (Remarks to the Author):

The authors presented a very interesting report on a CVD process assisted by water vapor that they called "hydroxide vapor phase epitaxy (OHVPE)" to produce high quality WS₂ monolayers. The authors study the samples using different characterization techniques such as Raman, PL, STM and fabricated devices to demonstrate the improvement of the electrical properties of OHVPE-grown samples compared to CVD-grown WS₂ that used the standard method with oxide precursors. Additionally, they perform kinetic simulations using the NEB method to compare the energy barriers for sulfurization in the presence of W-OH and W-O edges. Although water assisted CVD have been already implemented to produce 2D TDMs and lateral heterostructures, the study presented here goes more in depth about how the crystal quality (in terms of density of defects) is improved by the presence of W-OH bonds. In my opinion, this is a very interesting study, and it will benefit a large community involved in the growth of 2D-materials. That is why I think the manuscript is suitable for publication in Nature Communications. However, there are some points that need further clarifications, see the comments below.

1- One of the main limitations in the CVD processes is the inhomogeneity of the samples due to temperature gradients of the furnace at the substrate's positions, as well as gradients of the gas precursors across the radial direction of the tube (cross-section). This limitation is already significant in 1 inch diameter quartz tube reactors and are expected to be even more pronounced in larger diameter tubes like the one used in this study (3 inches). However, in the manuscript there is only one figure (Fig. 1c) that shows one individual island in an area smaller than 70 x 70 microns². Since the authors claim that using this method it is possible to grow WS₂ monolayers covering a 2 inches diameter substrate (Fig.1e), it is important that the authors present more evidence of sample homogeneity. Figure 1e is not enough evidence, a picture of any sapphire substrate (even without growth) will look like figure 1e. The authors should provide a low magnification image (approximately 2 x 2 mm, or larger) showing the island distribution, the density and thickness homogeneity.

2- The statistics in Figures 2b and c, is given in probability on the y-axis, it would be more informative to give it in (counts), so the reader can have a better idea of how many data points were used in the histograms. Was this statistic performed in different crystals and samples like the extended data figure 2b? or it was taken from one sample only?

3- To show sample homogeneity over a large area, Raman and PL (position and intensity) maps should be used to demonstrate the spatial quality distribution of the monolayers, either in a collection of islands and/or in continuous films. For instance, a map of the A_{1g} peak width, and a map of the LA(M) relative intensity. On the other hand, for thickness homogeneity, a map showing the intensity ratio between the 2LA(M)/A_{1g} will be enough to show thickness distribution since according to figure 2a, for 532 nm excitation laser, the 2LA(M) is still close to the double resonance that makes this mode more intense (in WS₂) under certain excitation wavelengths [Scientific Reports Vol. 3, Article number: 1755 (2013)].

More details on the growth process are needed if other groups are expected to reproduce these experiments, for instance:

4- What is the temperature gradient at the substrate position? The authors mentioned a substrate temperature range between 800-950C, this is a considerably large range; a significant variation in the sample quality, morphology, thickness, and homogeneity could be expected at each position within this gradient; this is especially important if a 2" diameter substrate (like the one in Fig. 1e) is used (this is linked to my first comment).

5- How was the water introduced? Was a bubbler used? If so, was it immersed in a thermal bath to keep a constant temperature? Variations on the water temperature can affect the concentration of water vapor in the carrier gas, did the authors observed any kind of fluctuations in the results due to these variations (if any)? How reproducible was the experiment? Did the authors attempt different conditions with different water vapor

concentration (i.e., different temperatures of the bubbler) in the carrier gas?

6- How was the cooling down performed? In the same carrier gas? Was the water vapor present during the cooling down? Cooling down temperature rate? etc.

Reviewer #2 (Remarks to the Author):

In this manuscript, the authors report that high-quality WS₂ monolayer can be realized by hydroxide vapor phase epitaxy. The defect density is one order of magnitude lower than that from conventional CVD methods. The bottom-gated FET devices exhibit high room-temperature mobility of ~ 198 cm²/Vs and high on-state current of ~ 400 μ A/ μ m, comparable to those from exfoliated flakes.

Overall, for monolayer TMDs based FET devices, such a room-temperature mobility is attractive. However, in my point of view, the data and discussion about the electrical characterization, the key point of this manuscript, are not sufficient at present version. It mainly includes: (1) As a new low-defect-density material system, authors should characterize more FETs instead of only nine shown in Figure 4c (e.g. more than 150 MoS₂ FETs are demonstrated in Nat. Nanotech. 2021, 16, 1201). Especially considering that the authors succeed in growing 2-inch OHVPE-WS₂ monolayer film, and the simplicity of FETs characterization. (2) The authors point out that the density of charged defects is a critical factor that limits the performance of 2D monolayers. Considering the intrinsic low-defect-density nature of OHVPE-WS₂ monolayer, whether it is possible to further increase the device performance by reducing the charge impurity density from WS₂-SiO₂ interfaces (e.g. using h-BN encapsulation) and lithographic processes. After addressing these concerns, I think the manuscript can be published in this journal. I also hope the authors could consider the following questions to make the manuscript better.

- (1) Gate hysteresis is crucial for judging the quality of 2D semiconductor FETs. Could authors provide the dual-sweep transfer curves, as well as output curves?
- (2) Why two different metal contacts (Bi and Al) were adopted for short-channel device and four-terminal device? Which metal contact is better for FET?
- (3) Could authors provide the mobility difference between two- and four-probe techniques?
- (4) The font size in Figure 2a is too small.

Reviewer #3 (Remarks to the Author):

In this study, the authors present a modified CVD approach to grow TMDCs monolayers, referred as "hydroxide vapor phase epitaxy". The authors compared the growth between these two approaches, "standard CVD" and "HVPE" by means of Raman, PL, transport measurements, STM and modeling of the chemical reaction kinetic barriers in each method. All these strongly suggest that the achieved monolayer TMDC by the HVPE are of higher optical and electrical properties with a directly measured lower density of defects. The kinetic simulation of the different chemical reactions in both cases suggests the kinetic barriers for the formation of the TMDC in HVPE are lower and thus favorable over the "standard CVD". The authors show that high optical and electrical grade material can be achieved. My comments:

1. The authors defined the procedure as "hydroxide vapor phase epitaxy", however, there is no proof to have epitaxial growth. The only figure showing more than a single triangular domain, extended data Fig 3, suggest there is no epitaxy, and if epitaxy is observed, how consistent is that? There should be very clear indications for epitaxial growth. Maybe the term should be modified to "hydroxide vapor phase growth" or "hydroxide chemical vapor deposition", etc.
2. One of the most important parameters in the growth in this case is the introduction of moisture. The technical details about that part is missing. "Moisture was delivered into

the growth chamber by Ar gas flow (180 s.c.c.m.) at atmospheric pressure.” Is not enough. What was the estimated moisture concentration/volume flow?

3. The introduction of water/moisture or oxygen during CVD/MOCVD is not new and was reported in the past, however not cited and mentioned in the manuscript. Here are some examples: 2017 2D Mater. 4 021024; ACS Materials Letters 2020 2 (1), 42-48; ACS Nano 2021, 15, 1, 526–538; J. Am. Chem. Soc. 2015, 137, 50, 15632–15635; ACS Nano 2017, 11, 12, 12001–12007.

4. References to the “defect-sensitive modes”, page 3, line 97, are missing.

To conclude, the authors present an improvement in the growth of TMDCs using a hydroxide-supported CVD approach. As mentioned above, the manuscript is missing important references directly related to the method. The terminology applied (“epitaxy”) is wrong, misunderstood or simply not proved. The work presents high quality experimental (STM, transport measurements, etc.) and simulation data.

REVIEWER COMMENTS & AUTHOR RESPONSES

*The responses are shown in blue fonts.

Reviewer #1 (Remarks to the Author):

The authors presented a very interesting report on a CVD process assisted by water vapor that they called “hydroxide vapor phase epitaxy (OHVPE)” to produce high quality WS₂ monolayers. The authors study the samples using different characterization techniques such as Raman, PL, STM and fabricated devices to demonstrate the improvement of the electrical properties of OHVPE-grown samples compared to CVD-grown WS₂ that used the standard method with oxide precursors. Additionally, they perform kinetic simulations using the NEB method to compare the energy barriers for sulfurization in the presence of W-OH and W-O edges. Although water assisted CVD have been already implemented to produce 2D TDMs and lateral heterostructures, the study presented here goes more in depth about how the crystal quality (in terms of density of defects) is improved by the presence of W-OH bonds. In my opinion, this is a very interesting study, and it will benefit a large community involved in the growth of 2D-materials. That is why I think the manuscript is suitable for publication in Nature Communications. However, there are some points that need further clarifications, see the comments below.

We are glad that the reviewer finds this work interesting and significant for the community. We have provided point-by-point answers to the comments and concerns raised by the reviewer.

1- One of the main limitations in the CVD processes is the inhomogeneity of the samples due to temperature gradients of the furnace at the substrate’s positions, as well as gradients of the gas precursors across the radial direction of the tube (cross-section). This limitation is already significant in 1 inch diameter quartz tube reactors and are expected to be even more pronounced in larger diameter tubes like the one used in this study (3 inches). However, in the manuscript there is only one figure (Fig. 1c) that shows one individual island in an area smaller than 70 x 70 microns². Since the authors claim that using this method it is possible to grow WS₂ monolayers covering a 2 inches diameter substrate (Fig.1e), it is important that the authors present more evidence of **sample homogeneity**. Figure 1e is not enough evidence, a picture of any sapphire substrate (even without growth) will look like figure 1e. The authors should provide a low magnification image (approximately 2 x 2 mm, or larger) showing the island distribution, the density and thickness homogeneity.

Response: We acknowledge the reviewer’s valuable comments. We fully agree that sample homogeneity is critical for TMDC growth. Thus, most of our results, including Raman, PL, STM, and electrical performance were collected from different positions and batches to make the statistical distribution.

To show the island distribution, we present in **Figure R1** the optical images ($1 \times 1 \text{ mm}^2$) taken from different positions of a WS_2 sample across the wafer before island merging, where the WS_2 grains evenly and densely disperse on the substrate. Note that the truncated triangle shape of deposited WS_2 monolayers is attributed to growth parameters, including high growth temperature and high step height of the atomic steps on the selected C/A sapphire with 1° miscut angle.

Furthermore, to present the sample homogeneity, PL and Raman mapping of $2 \times 2 \text{ mm}^2$ WS_2 film were performed as shown in **Figure R2**. By using a 488 nm excitation laser, the E_{2g} and A_{1g} Raman modes of WS_2 are easier to clarify during the mapping process. As shown in **Figures R2b and c**, the average intensity ratio of $I_{E_{2g}}/I_{A_{1g}}$ is 0.7, and the average frequencies difference between E_{2g} and A_{1g} is 61.5 cm^{-1} which corresponds to WS_2 monolayer thickness¹. The PL intensity and peak position mapping images (**Figure R2d and e**) show that the average emission peak position of the film is around 620 nm (2 eV) with identical PL intensity. It is hard to avoid seeds and multilayer growth in the large-scale deposition process (**Figure R2f**), so a small quantity of brighter or darker dots can be found in the mapping image. Similar to other reports², a higher density of multilayers and seeds would be found at the center of the film compared to the edge. Further study is needed to decrease these seeds and multilayers.

Action: We have added the following sentence to the main text on page 3: “*PL and Raman mapping results in Supporting Information Fig.S3 present a homogeneous and high-quality OHVPD- WS_2 film*” The corresponding text and figures were updated to **Supporting Information Note S2 and Fig. S3**.

Figure R1 Four $1 \times 1 \text{ mm}$ optical images show the island distribution of OHVPD- WS_2 .

Figure R2 Optical image (a), Raman mapping results (b, c), and PL mapping results (d, e) of 2 x 2 mm² OHVPD-WS₂ film. (f) Optical image shows small quantities of WS₂ seeds and multilayers.

2- The statistics in Figures 2b and c, is given in probability on the y-axis, it would be more informative to give it in (counts), so the reader can have a better idea of how many data points were used in the histograms. Was this statistic performed in different crystals and samples like the extended data figure 2b? or it was taken from one sample only?

Response: We appreciated the reviewer’s constructive suggestions. We have modified the statistical Raman data as shown in **Figure R3** with “counts” on the y-axis. 50 data points for each type were collected from 3~4 sample batches, and every data point was taken from different crystals.

Action: We have added the following sentence to the main text on page 4: “*To qualitatively compare the defect level, 50 Raman spectra from various single crystals were collected for each type of samples .*” In the meantime, **Figures 2b and c** have been updated.

Figure R3 Statistic distribution of A_{1g} peak width (a) and LA(M) normalized intensity (b) for OHVPD- and CVD- WS_2 monolayers.

3- To show sample homogeneity over a large area, Raman and PL (position and intensity) maps should be used to demonstrate the spatial quality distribution of the monolayers, either in a collection of islands and/or in continuous films. For instance, a map of the A_{1g} peak width, and a map of the LA(M) relative intensity. On the other hand, for thickness homogeneity, a map showing the intensity ratio between the $2LA(M)/A_{1g}$ will be enough to show thickness distribution since according to figure 2a, for 532 nm excitation laser, the $2LA(M)$ is still close to the double resonance that makes this mode more intense (in WS_2) under certain excitation wavelengths [Scientific Reports Vol. 3, Article number: 1755 (2013)].

Response& Action: Please refer to the response to comment #1. Raman and PL maps of the continuous film were present in **Figure R2**. Note that 488 nm excitation laser was applied since 532 nm excitation causes the weak intensity of LA(M) and A_{1g} Raman signals with poor resolution. The statistical results in Figure 2 were collected and fitted one by one to make the data more reliable. According to the reference¹, 488 nm excitation laser could also be used to distinguish the layer numbers of WS_2 by $I_{E_{2g}}/I_{A_{1g}}$ and $A_{1g} - E_{2g}$. The related figure has been updated to **Supporting Information Fig. S3**.

More details on the growth process are needed if other groups are expected to reproduce these experiments, for instance:

4- What is the temperature gradient at the substrate position? The authors mentioned a substrate temperature range between 800-950C, this is a considerably large range; a significant variation in the sample quality, morphology, thickness, and homogeneity could be expected at each position within this gradient; this is especially important if a 2" diameter substrate (like the one in Fig. 1e) is used (this is linked to my first comment).

Response: We thank the reviewer for raising this important question. **Figure R4a** shows the temperature gradient of our single-heating zone furnace when the heating temperature is set at 1050°C. The position marked as 0 cm corresponds to the center of the furnace. The

substrates were distanced from the center by 9 to 14 cm (the temperature range is around 950-800°C). We agree with the reviewer that such a temperature range caused some sample growth variations. **Figure R4b** presents the typical PL peaks of OHVPD-WS₂ collected from different temperature regions. It is evident that the PL peaks of WS₂ film had a redshift when the substrate was put in a higher temperature region (region A). Given that very limited defective and doping variations are concluded from the low-temperature PL and STM results, the possible explanation is that the OHVPD-WS₂ film grown in a higher temperature zone contains higher tensile strain, which results in a PL peak redshift.³ Although the thermal expansion coefficient difference between WS₂ and sapphire substrate is small,⁴ a higher deposition temperature (like 950°C) still causes a higher strain effect on the deposited film. Thus, in future work, we will consider using cold-wall and susceptor (heater)-type systems to avoid a large temperature gradient in a tube furnace.

On the other hand, as concluded in response to Comment #1, the temperature gradient does not lead to noticeable variation in morphology and thickness of samples because the OHVPD method provides more volatile W-OH reactants that are not hampered by large temperature drops.

Action: We have added the following sentence to the Methods on page 12: “*More details on the growth process and results are provided in Supporting Information Note S3*”. The corresponding text and figures were updated to **Supporting Information Note S3 and Fig. S7**.

Figure R4 (a) Temperature profile of the single-heating zone furnace. (Setting temperature is 1050) (b) PL spectrums of OHVPD-WS₂ film from different regions.

5- How was the water introduced? Was a bubbler used? If so, was it immersed in a thermal bath to keep a constant temperature? Variations on the water temperature can affect the concentration of water vapor in the carrier gas, did the authors observed any kind of fluctuations in the results due to these variations (if any)? How reproducible was the experiment? Did the authors attempt different conditions with different water vapor concentration (i.e., different temperatures of the bubbler) in the carrier gas?

Response: Yes, we adopted a bubbler setup to introduce the water vapor into the growth chamber, as shown in **Figure R5a**. During the experiment, we fixed the carrier gas flow of Ar and adjusted the water temperature using a thermal bath to change the water vapor concentration. The Antoine equation⁵ is able to estimate the partial pressure of water (p_{H_2O}); hence, the variable of water vapor in our experiment can be well gauged. The provided analyses of spectroscopic and devices from various batches have also proved good reproducibility of the growth.

High p_{H_2O} (92.59 torr, 50°C) results in over oxidization during the reaction, where excess oxides can be found on as-grown samples (**Figure R5b**). In contrast, low p_{H_2O} (17.54 torr, 20°C) lead to a low deposition with small grains due to scarcity of metal supply (**Figure 5c**). The optimized water bath temperature for WS₂ deposition is 35°C, which provides p_{H_2O} = 42.20 torr (**Figure R5d**).

Action: The corresponding text and figures were updated to **Supporting Information Note S3 and Fig. S8**.

Figure R5 (a) Schematic illustration of our bubbler setup for water vapor supply. Optical images of OHVPD-WS₂ growth results under (b) high p_{H_2O} (92.59 torr, 50°C), (c) low p_{H_2O} (17.54 torr, 20°C), and (d) optimized p_{H_2O} (42.20 torr, 35°C).

6- How was the cooling down performed? In the same carrier gas? Was the water vapor present during the cooling down? Cooling down temperature rate? etc.

Response: We applied natural cooling (~30 min to lower than 200°C) under the same carrier gas (Ar/H₂ 180/20 s.c.c.m.) without the presence of water vapors.

Action: We have added the following sentence to the Methods on page 12: “The growth was kept for 15 min and followed by natural cooling to room temperature with the same carrier gas (Ar/H₂ 180/20 s.c.c.m.) without the presence of water vapors.”

Reviewer #2 (Remarks to the Author):

In this manuscript, the authors report that high-quality WS₂ monolayer can be realized by hydroxide vapor phase epitaxy. The defect density is one order of magnitude lower than that from conventional CVD methods. The bottom-gated FET devices exhibit high room-temperature mobility of ~198 cm²/Vs and high on-state current of ~400 μA/μm, comparable to those from exfoliated flakes.

Overall, for monolayer TMDs based FET devices, such a room-temperature mobility is attractive. However, in my point of view, the data and discussion about the electrical characterization, the key point of this manuscript, are not sufficient at present version. It mainly includes:

We appreciate that the reviewer regards our results are attractive. A point-by-point response to the comments and concerns raised can be found below.

(1) As a new low-defect-density material system, authors should characterize more FETs instead of only nine shown in Figure 4c (e.g. more than 150 MoS₂ FETs are demonstrated in Nat. Nanotech. 2021, 16, 1201). Especially considering that the authors succeed in growing 2-inch OHVPE-WS₂ monolayer film, and the simplicity of FETs characterization.

Response: We thank the reviewer for the suggestion. Contrary to the reported approach for wafer-scale single-crystalline TMDC films⁶, which proves high uniformity using batch-produced FET arrays, the topic elaborated in this work is the superiority of the OHVPD method in preparing low-defect TMDC monolayers. Hence, we individually fabricated the devices for each crystal to preclude transport through grain boundary for a better evaluation of sample quality. With the short period of revision time and our clean opening time constraints imposed by the COVID-19, we tried our best to add more FET results (up to 15 of four-probe measurement results and 35 of two-probe measurement results) into **Figure 4c and Supporting Information Fig. S5, as shown in Figure R6.**

Action: The related data and figures have been added in **Supporting Information Fig. S5 and Figure 4.**

Figure R6 (a) Benchmarking field-effect mobility for WS₂ monolayers based on four-probe measurements. (b) Benchmarking field-effect mobility for WS₂ monolayers based on two-probe measurements. The points with centerline interior represent the mean value.

(2) The authors point out that the density of charged defects is a critical factor that limits the performance of 2D monolayers. Considering the intrinsic low-defect-density nature of OHVPE-WS₂ monolayer, whether it is possible to further increase the device performance by reducing the charge impurity density from WS₂-SiO₂ interfaces (e.g. using h-BN encapsulation) and lithographic processes. After addressing these concerns, I think the manuscript can be published in this journal.

Response: We consent to the reviewer's statement that the passivation of the channel/dielectric interface and improved lithographic processes shall enhance the device performance further. Many prior research works have already demonstrated the performance enhancement by encapsulation of 2D materials by h-BN⁷ or organic molecule⁸.

However, the addition of the low-dielectric-constant hBN layers actually degrades the gating efficiency in transistors.⁹ Meanwhile, we have to admit that we have run out the high quality hBN crystals and we could not get the crystal and fabricate the devices in a short timeframe.

Our device structure comprising scalable high-k materials deposited by ALD has already proved superior quality compared with conventional CVD. We hope to leave the further engineering tasks to improve the device performance to another work that will adopt a susceptor-type of growth tool for the single crystal growth of 2D using OHVPD.

I also hope the authors could consider the following questions to make the manuscript better. (1) Gate hysteresis is crucial for judging the quality of 2D semiconductor FETs. Could authors provide the dual-sweep transfer curves, as well as output curves?

Response: As requested, **Figure R7** shows the typical dual-sweep transfer curves of our devices. The normalized hysteresis width is 40 mV/MV cm⁻¹, which is on a par with reported

hysteresis values and indicates the presence of low border traps and interface states. Furthermore, the output curve (**Figure R8**) of the short-gate-length (100 nm) FET is provided, which demonstrates promising current control and saturation.

Action: For the hysteresis characteristics, we have added the following sentence to the main text on page 7: “*Supporting Information Fig. S4 shows the typical dual-sweep transfer curves of our devices. The normalized hysteresis width is 40 mV/MV cm^{-1} , which is on a par with reported hysteresis values and indicates the presence of low border traps and interface states.*” In the meantime, **Fig. S4** has been added to the Supporting Information.

For the output characteristics, we have added the following sentence to the main text on page 8: “*Supporting Information Fig. S6 shows the output characteristics of this short-gate-length FET, which demonstrates promising current control and saturation.*” In the meantime, **Fig. S6** has been added to the Supporting Information.

Figure R7 Typical dual-sweep transfer curve characteristics of OHVPD-WS₂ monolayer device.

Figure R8 The output characteristics of the short-gate-length (100 nm) FET based on OHVPD-WS₂.

(2) Why two different metal contacts (Bi and Al) were adopted for short-channel device and four-terminal device? Which metal contact is better for FET?

Response: Using Bi contact is referred according to our previous work¹⁰, where the semimetal Bi can avoid the formation of metal-induced gap states to achieve contact barrier-free in MoS₂ n-FETs. Thus, we adopted Bi contact to demonstrate high ON current density in short-channel WS₂ FETs.

On the other hand, Al is recognized as one of the favorable metal contacts for WS₂ n-FETs because of their comparable work function (Al~ 4.1 eV)⁷. Most importantly, Al-WS₂ contact is relatively more stable. For the four-probe measurement, we normally need to reserve the measurement systems and the waiting time was long. Therefore, Al was adopted as the metal for extracting the value of field-effect mobility to compare with other reported results.

Bi metal typically leads to high On current for the 2D FETs, but the devices may need to be properly passivated in future applications.

(3) Could authors provide the mobility difference between two- and four-probe techniques?

Response: As requested, **Figure R9** shows the field-effect mobility difference between two- and four-probe measurements for the same device. Similar to the previous report¹¹, the two-probe mobility is much lower than four-probe mobility due to the considerable contact

resistance influence. Thus, we applied four-probe measurements for fairly comparing the sample quality.

Figure R9 (a) Two-probe transfer characteristic of the monolayer OHVPD-WS₂ field-effect transistor with $\mu = 33 \text{ cm}^2\text{V}^{-1}\text{s}^{-1}$. (b) Four-probe transfer characteristic of the monolayer OHVPD-WS₂ field-effect transistor with $\mu = 198 \text{ cm}^2\text{V}^{-1}\text{s}^{-1}$.

(4) The font size in Figure 2a is too small.

Response& Action: We appreciated the reviewer’s careful reading and useful suggestions. We have enlarged the font size in **Figure 2a**.

Reviewer #3 (Remarks to the Author):

In this study, the authors present a modified CVD approach to grow TMDCs monolayers, referred as “hydroxide vapor phase epitaxy”. The authors compared the growth between these two approaches, “standard CVD” and “HVPE” by means of Raman, PL, transport measurements, STM and modeling of the chemical reaction kinetic barriers in each method. All these strongly suggest that the achieved monolayer TMDC by the HVPE are of higher optical and electrical properties with a directly measured lower density of defects. The kinetic simulation of the different chemical reactions in both cases suggests the kinetic barriers for the formation of the TMDC in HVPE are lower and thus favorable over the “standard CVD”. The authors show that high optical and electrical grade material can be achieved. My comments:

We appreciate that the reviewer recognizes our high optical and electrical grade materials. A point-by-point response to the comments can be found below.

1. The authors defined the procedure as “hydroxide vapor phase epitaxy”; however, there is no proof to have epitaxial growth. The only figure showing more than a single triangular domain, extended data Fig 3, suggest there is no epitaxy, and if epitaxy is observed, how

consistent is that? There should be very clear indications for epitaxial growth. Maybe the term should be modified to “hydroxide vapor phase growth” or “hydroxide chemical vapor deposition”, etc.

Response: We have changed the “hydroxide vapor phase epitaxy” to “hydroxide vapor phase deposition,” and we have changed all the abbreviations to OHVPD. In this study, we focus on the chemistry of the growth rather than the substrate control for epitaxy. We will start a separate project to combine the growth with substrate engineering to realize the epitaxy. We thank the reviewer for pointing this out.

2. One of the most important parameters in the growth in this case is the introduction of moisture. The technical details about that part is missing. “Moisture was delivered into the growth chamber by Ar gas flow (180 s.c.c.m.) at atmospheric pressure.” Is not enough. What was the estimated moisture concentration/volume flow?

Response: Many thanks for raising this important question. Figure R5a shows the bubbler setup we used to introduce the moisture. During the experiment, we fixed the carrier gas flow of Ar and adjusted the water temperature using a thermal bath to change the water vapor concentration. The Antoine equation⁵ is able to estimate the partial pressure of water (p_{H_2O}). Figure R5b, c, and d show the growth results with various p_{H_2O} , and the optimized temperature for WS₂ deposition is 35°C which provides $p_{H_2O} = 42.20$ torr.

Action: We have updated the moisture delivery setup and details in the **Supporting Information Note S4 and Fig. S8**.

Figure R5 (a) Schematic illustration of our bubbler set up for water vapor supply. Optical images of OHVPD-WS₂ growth results under (b) high p_{H_2O} (92.59 torr, 50°C), (c) low p_{H_2O} (17.54 torr, 20°C), and (d) optimized p_{H_2O} (42.20 torr, 35°C).

3. The introduction of water/moisture or oxygen during CVD/MOCVD is not new and was reported in the past, however not cited and mentioned in the manuscript. Here are some examples: 2017 2D Mater. 4 021024; ACS Materials Letters 2020 2 (1), 42-48; ACS Nano 2021, 15, 1, 526–538; J. Am. Chem. Soc. 2015, 137, 50, 15632–15635; ACS Nano 2017, 11, 12, 12001–12007.

Response & Action: We appreciate the reviewer’s suggestion. We have added the sentence “Transport agents like water¹²⁻¹⁴ and oxygen^{15,16} have been used to enhance the volatilization of metal source for improving the growth; however, the impact on materials have seldom been explored.” to the introduction part on Page 2. The relevant references have been added to the manuscript.

4. References to the “defect-sensitive modes”, page 3, line 97, are missing.

Response& Action: We appreciated the reviewer’s careful reading. We have added Ref [17] and Ref [18] to the sentence on page 3.

To conclude, the authors present an improvement in the growth of TMDCs using a hydroxide-supported CVD approach. As mentioned above, the manuscript is missing important references directly related to the method. The terminology applied (“epitaxy”) is wrong, misunderstood or simply not proved. The work presents high quality experimental (STM, transport measurements, etc.) and simulation data.

We appreciate that the reviewer regards our experimental and simulation data are high quality. Also, we are glad that the reviewer acknowledges the improvement of this approach. We have modified the manuscript according to the reviewers’ constructive suggestions.

References

- 1 Berkdemir, A. *et al.* Identification of individual and few layers of WS₂ using Raman Spectroscopy. *Scientific Reports* **3**, 1755, doi:10.1038/srep01755 (2013).
- 2 Sebastian, A., Pendurthi, R., Choudhury, T. H., Redwing, J. M. & Das, S. Benchmarking monolayer MoS₂ and WS₂ field-effect transistors. *Nature Communications* **12**, 693, doi:10.1038/s41467-020-20732-w (2021).
- 3 Chang, C.-H., Fan, X., Lin, S.-H. & Kuo, J.-L. Orbital analysis of electronic structure and phonon dispersion in MoS₂, MoSe₂, WS₂, and WSe₂ monolayers under strain. *Physical Review B* **88**, 195420, doi:10.1103/PhysRevB.88.195420 (2013).
- 4 McCreary, K. M. *et al.* The Effect of Preparation Conditions on Raman and Photoluminescence of Monolayer WS₂. *Scientific Reports* **6**, 35154, doi:10.1038/srep35154 (2016).
- 5 Lide, D. R. *CRC handbook of chemistry and physics*. Vol. 85 (CRC press, 2004).
- 6 Li, T. *et al.* Epitaxial growth of wafer-scale molybdenum disulfide semiconductor single crystals on sapphire. *Nature Nanotechnology* **16**, 1201-1207, doi:10.1038/s41565-021-00963-8 (2021).
- 7 Iqbal, M. W. *et al.* High-mobility and air-stable single-layer WS₂ field-effect transistors sandwiched between chemical vapor deposition-grown hexagonal BN films. *Scientific Reports* **5**, 10699, doi:10.1038/srep10699 (2015).
- 8 Yu, Z. *et al.* Towards intrinsic charge transport in monolayer molybdenum disulfide by defect and interface engineering. *Nature Communications* **5**, 5290, doi:10.1038/ncomms6290 (2014).
- 9 Knobloch, T. *et al.* The performance limits of hexagonal boron nitride as an insulator for scaled CMOS devices based on two-dimensional materials. *Nature Electronics* **4**, 98-108, doi:10.1038/s41928-020-00529-x (2021).
- 10 Shen, P.-C. *et al.* Ultralow contact resistance between semimetal and monolayer semiconductors. *Nature* **593**, 211-217, doi:10.1038/s41586-021-03472-9 (2021).
- 11 Nazir, G., Khan, M. F., Iermolenko, V. M. & Eom, J. Two- and four-probe field-effect and Hall mobilities in transition metal dichalcogenide field-effect transistors. *RSC Advances* **6**, 60787-60793, doi:10.1039/C6RA14638D (2016).

- 12 Kastl, C. *et al.* The important role of water in growth of monolayer transition metal
dichalcogenides. *2D Materials* **4**, 021024, doi:10.1088/2053-1583/aa5f4d (2017).
- 13 Zhao, Y. & Jin, S. Controllable Water Vapor Assisted Chemical Vapor Transport Synthesis of
WS₂–MoS₂ Heterostructure. *ACS Materials Letters* **2**, 42-48,
doi:10.1021/acsmaterialslett.9b00415 (2020).
- 14 Cohen, A. *et al.* Growth-Etch Metal–Organic Chemical Vapor Deposition Approach of WS₂
Atomic Layers. *ACS Nano* **15**, 526-538, doi:10.1021/acsnano.0c05394 (2021).
- 15 Chen, W. *et al.* Oxygen-Assisted Chemical Vapor Deposition Growth of Large Single-Crystal
and High-Quality Monolayer MoS₂. *Journal of the American Chemical Society* **137**, 15632-
15635, doi:10.1021/jacs.5b10519 (2015).
- 16 Yu, H. *et al.* Wafer-Scale Growth and Transfer of Highly-Oriented Monolayer MoS₂
Continuous Films. *ACS Nano* **11**, 12001-12007, doi:10.1021/acsnano.7b03819 (2017).
- 17 Li, J. *et al.* Atypical Defect-Mediated Photoluminescence and Resonance Raman
Spectroscopy of Monolayer WS₂. *The Journal of Physical Chemistry C* **123**, 3900-3907,
doi:10.1021/acs.jpcc.8b11647 (2019).
- 18 Shi, W. *et al.* Raman and photoluminescence spectra of two-dimensional nanocrystallites of
monolayer WS₂ and WSe₂. *2D Materials* **3**, 025016,
doi:10.1088/2053-1583/3/2/025016 (2016).

REVIEWERS' COMMENTS

Reviewer #1 (Remarks to the Author):

The authors have addressed my major concerns. In my opinion the reviewed version of the manuscript is now acceptable for publication.

Reviewer #2 (Remarks to the Author):

After reading the revised manuscript and the authors' responses to the referees, I believe the manuscript can be published in Nature Communications.

Reviewer #3 (Remarks to the Author):

I believe the authors successfully answer mine and the other reviewer's concerns, and therefore, recommend to publish the manuscript in Nature Communications.

One minor thing, the authors probably missed is that they did not add the references they claimed in the rebuttal letter on the "defect-sensitive modes", page 3, line 97.

REVIEWER COMMENTS & AUTHOR RESPONSES

*The responses are shown in blue fonts.

Reviewer #1 (Remarks to the Author):

The authors have addressed my major concerns. In my opinion the reviewed version of the manuscript is now acceptable for publication.

Response: We appreciate that the reviewer recognizes our manuscript is acceptable for publication.

Reviewer #2 (Remarks to the Author):

After reading the revised manuscript and the authors' responses to the referees, I believe the manuscript can be published in Nature Communications.

Response: We are glad that the reviewer recognizes our manuscript can be published in Nature Communications. Thanks a lot.

Reviewer #3 (Remarks to the Author):

I believe the authors successfully answer mine and the other reviewer's concerns, and therefore, recommend to publish the manuscript in Nature Communications.

One minor thing, the authors probably missed is that they did not add the references they claimed in the rebuttal letter on the “defect-sensitive modes”, page 3, line 97.

Response: We acknowledge the reviewer’s recommendation of publishing our manuscript in Nature Communications.

In previous version, we put the reference at the end of that sentence which may cause the confusion. Thus, we move the related reference to the end of “defect-sensitive modes” to make it clear.

The related reference is [Li, J. et al. Atypical Defect-Mediated Photoluminescence and Resonance Raman Spectroscopy of Monolayer WS₂. The Journal of Physical Chemistry C 123, 3900-3907, doi:10.1021/acs.jpcc.8b11647 (2019).]